# RIFIS: A Novel Rice Field Sidewalk Detection Dataset for Walk-Behind Hand Tractor

Padma Nyoman Crisnapati [ID] and Dechrit Maneetham *

Department of Mechatronics Engineering, Rajamangala University of Technology Thanyaburi, Pathum Thani 12110, Thailand
* Correspondence: dechrit_m@rmutt.ac.th

**Abstract:** Rice field sidewalk (RIFIS) identification plays a crucial role in enhancing the performance of agricultural computer applications, especially for rice farming, by dividing the image into areas of rice fields to be ploughed and the areas outside of rice fields. This division isolates the desired area and reduces computational costs for processing RIFIS detection in the automation of ploughing fields using hand tractors. Testing and evaluating the performance of the RIFIS detection method requires a collection of image data that includes various features of the rice field environment. However, the available agricultural image datasets focus only on rice plants and their diseases; a dataset that explicitly provides RIFIS imagery has not been found. This study presents an RIFIS image dataset that addresses this deficiency by including specific linear characteristics. In Bali, Indonesia, two geographically separated rice fields were selected. The initial data collected were from several videos, which were then converted into image sequences. Manual RIFIS annotations were applied to the image. This research produced a dataset consisting of 970 high-definition RGB images (1920 × 1080 pixels) and corresponding annotations. This dataset has a combination of 19 different features. By utilizing our dataset for detection, it can be applied not only for the time of rice planting but also for the time of rice harvest, and our dataset can be used for a variety of applications throughout the entire year.

**Dataset:** https://doi.org/10.21227/pnxx-3t40.

**Dataset License:** CC-BY 4.0.

**Keywords:** rice field; sidewalk detection; dataset; walk-behind hand tractor; Mask R-CNN

## 1. Introduction

The rice field sidewalk line is a thin boundary that becomes the boundary of a rice field in Indonesia. This part assists in isolating the observed rice field region, which can ultimately be used as a computational reference for image processing for tractor automation, particularly in the plowing process. Consequently, rice field sidewalk identification is a key function in agricultural computer applications for tractor [1,2] navigation, UGV [3,4], monitoring [5], object detection [6], tracking [7], distance calculation [8], collision avoidance [9,10], and path planning [11]. Rice field sidewalk detection is a challenging task. A rice field scene's abundance of elements contributes to its complexity. Strongly linear foreground or background objects and environmental variables are prominently featured. Grass, soil, puddles, clouds, paddy field structures, and background landscapes are strong sources of linear features. Rice field sidewalk (RIFIS) partial occlusion is possible because the horizon line may not traverse the entire width of the image, and its visibility is localized to a small section or region of the image.

This scenario presents an additional difficulty for RIFIS detection methods based on projection-based computer vision, as they seek the presence of linear features in an image

by employing edge detection methods and linear transformation. Variable illumination, grass, puddles, and the resemblance between the rice field region and the sidewalk present an additional obstacle for the RIFIS detection algorithm. Depending on the level of gloss and glare of the water surface in the rice fields, there may be a slight color variation between the sidewalk and the rice fields. Moreover, atmospheric conditions can alter the hue of puddles. The current scenario presents a difficulty for the RIFIS detection approach, which attempts to distinguish sidewalks from rice fields through image processing.

For testing and performance evaluation, the method that seeks to address the problem of RIFIS identification requires collecting benchmark image data of rice fields. The dataset is the sole benchmark for evaluating the robustness of a procedure. Numerous datasets of rice field imaging have been offered by researchers; however, limitations to seedlings [12], disease [13–15], height [16], varieties [17], growth [18,19], pests [20] rather than rice, the absence of background objects, low-resolution photos, and the lack of an RIFIS in this collection leave room for development. This research offered an RIFIS image dataset that satisfied the requirement by including distinct RIFIS characteristics in ploughing fields using hand tractors. The dataset primarily focused on computer vision and deep-learning-based RIFIS detection techniques. The entirety of the dataset was comprised of 18 videos, 3723 high-definition RGB images (1920 × 1080 pixels), and 970 labeled images. These images combined nineteen distinct characteristics for testing and evaluating the RIFIS detection algorithm. As an evaluation of the developed RIFIS dataset, Mask R-CNN was used as validation. This Mask R-CNN model was used because of its popularity in detecting various objects [21–24]. According to our knowledge, no other publicly available dataset currently contains images of these RIFISs.

### 1.1. Related Work

In this study, we reviewed the publication of publicly available rice field image datasets. These datasets include [12–15,25,26] with details that can be seen in Table 1. This section presents the purpose, attributes, and differences between these datasets and the dataset we collected. In [25], high-resolution image-based deep learning approaches were used to panicle datasets. The semi-supervised deep learning model training procedure was performed to annotate and modify the training dataset. Regarding the UAV seedling dataset [12], this research was focused on the annotation of the UAV picture dataset. The dataset was obtained using a UAV with many rotors that flew over rice fields to collect data. In addition, semi-automatic annotations were introduced to provide training data for rice seedling detection. Regarding the rice ear dataset [26], this research provided a dataset of 3300 rice ear samples that illustrated a variety of complex conditions, such as variable light and complex backgrounds, and rice and leaves that overlap. The acquired photos were manually tagged, and a data improvement technique was employed to expand the sample size. The researchers in [14] examined six major rice cultivars. The rice disease database contained images of rice leaves collected from the planting area's farms. The pictures were taken under an unmanaged natural environment. An RGB camera [13] was used to capture leaf disease picture data from rice plants. This study was conducted in the Mekong delta (VMD) rice fields in Vietnam. The study in [15] was also concerned with detecting rice illnesses. A DSLR camera was used to collect 1200 experimental photographs from a rice farm located on the University of Agricultural Sciences (UAS) campus in Dharwad, India. There were 750 photos showing rice fields impacted by fungal diseases, 250 images showing rice fields affected by bacterial diseases, and 200 images showing rice fields affected by viral diseases in the retrieved dataset. However, the field picture dataset initially collected with 1200 labeled photos was expanded to 12,000 labeled images by using several image enhancement methods. To our knowledge, however, the publicly available picture datasets for rice fields are restricted, and no RIFIS detection is available. To address this issue, we suggested creating a dataset of rice field sidewalk images named RIFIS.

**Table 1.** Summary of previous research datasets on rice fields.

| Title | Targeted Domain | Annotation Type | Number of Data | Place |
|---|---|---|---|---|
| Paddy Rice Imagery Dataset for Panicle Segmentation (2021) [25] | Panicle detection and segmentation tasks | Polygon | 400 images | Hokkaido University, Sapporo, Japan |
| A UAV Open Dataset of Rice Paddies for Deep Learning Practice (2021) [12] | Rice seedling detection | Bounding boxes | Rice seedling—28,047 images, Arable land—26,581 images | Wufeng District, Taichung, Taiwan |
| Rice Ear Counting Based on Image Segmentation and Establishment of a Dataset (2021) [26] | Rice ear detection | Polygon | 3300 images (originally 1100 images before augmentation) | Sichuan Agricultural University, Ya'an City, Sichuan Province, China |
| Classification of Rice Diseases using Convolutional Neural Network Models (2022) [15] | Rice disease detection | Bounding boxes | 12,000 images (originally 1200 images before augmentation) | University of Agricultural Sciences (UAS), Dharwad, India |
| Real-Time Disease Detection in Rice Fields in the Vietnamese Mekong Delta (2020) [13] | Rice disease detection | Bounding boxes | 116 images | Vietnamese Mekong Delta |
| Using Deep Learning Techniques to Detect Rice Diseases from Images of Rice Fields (2020) [14] | Rice disease detection | Polygon | 6300 images | Thailand |
| Proposed Rice Field Sidewalk (RIFIS) (2022) | Rice field sidewalk | Bounding boxes and Polygon | 3723 images and 18 videos | Denpasar, Bali, Indonesia |

*1.2. Research Contribution*

The salient contributions of this dataset were (1) it was the first novel dataset for the detection of the RIFIS in a two-wheeled hand tractor; (2) the diversity of features related to the foreground and background objects, state of the fields, level of illumination, luster, glare, standing water, cloud cover, and hand tractor movement. The proposed dataset presented RIFIS images collected using a tractor movement scenario with a spiral pattern in two separate locations in the province of Bali, Indonesia. Based on our knowledge, no other RIFIS image dataset is currently available. In addition to datasets in the form of videos and images, we also collected the location data (GPS) and orientation (accelerometer, gyroscope, and compass) of tractors during the ploughing process using the internet of things (IoT) technology.

## 2. Dataset Description

*2.1. Rice Field Sidewalk Dataset*

The RIFIS dataset presented in this work consisted of 16 videos with the size of 48.7 GB and 970 high-definition RGB images (1920 × 1080 pixels) and their annotations. Since the acquired raw material was a 1920 × 1080 pixel high-definition video, it was possible to extract several image sequences from a single video. By using this method, 24 images were recovered from each second of the raw video. The extracted images were named by concatenating the raw video source name and a postfix value that specified the order in which the images were extracted in the order in which they were made. For example, a raw video named "GH010327.MP4" (Figure 1a) was extracted into several image sequences starting at "GH010327_0100000.PNG". After that, several images were selected to be annotated and were given a name starting from "Sequence 0100000.JPG" (Figure 1b).

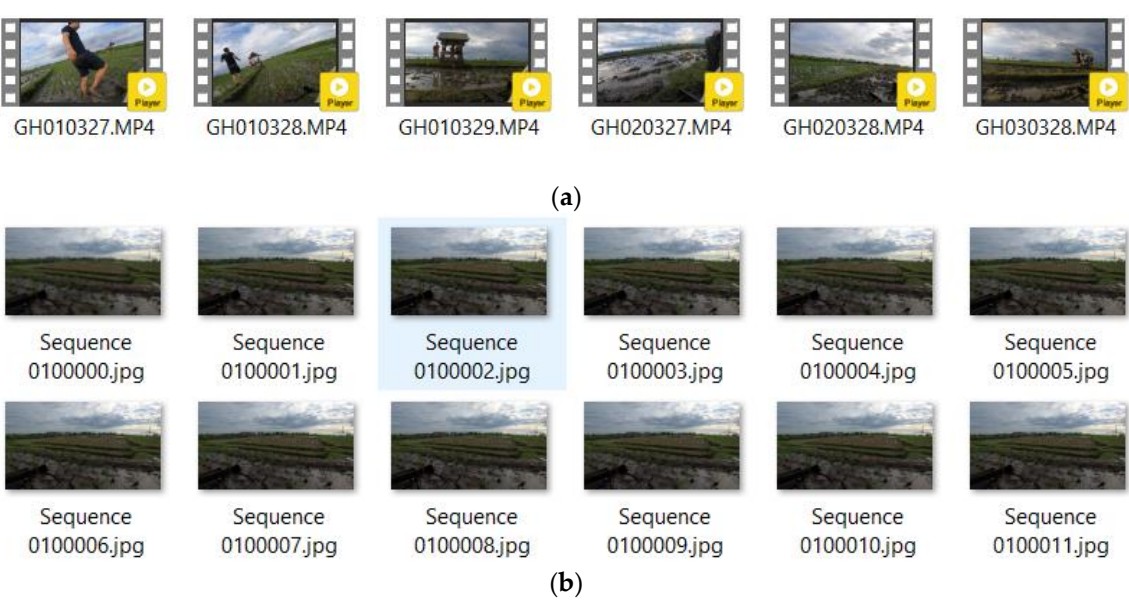

**Figure 1.** The collection process of the image sequences from video: (**a**) video data; (**b**) JPG image sequence.

Recognizing the surrounding environment was one of the requirements so that the tractor could recognize the inside and outside areas of the rice field. The easiest way to divide these two conditions was to detect the RIFIS. Based on the collected video dataset, observations were made on the environmental conditions of the rice fields to obtain several features that could be used. The dataset had 19 unique features. Combining day circumstances, weather and ambient factors, paddy fields, partial occlusion, foreground objects, and backdrops provided difficulties for the RIFIS detection algorithm. These 19 characteristics are categorized in Table 2. The data collection was only carried out in the afternoon due to limited available funds, so land leases, cameras, tractors, operators, etc., had limitations.

**Table 2.** RIFIS dataset features.

| Day Condition | Weather Condition | Rice Field State | Environmental Condition | Occlusion | Presence of Object |
|---|---|---|---|---|---|
| 1. Afternoon; | 2. Partially Cloudy; | 3. Partially Covered by Grass; 4. Watery; 5. Partially Ploughed; | 6. Mild to Strong Glare; 7. Variation in Rice Field Surface Color; 8. Not Smooth Color Transition Between Sidewalk and Rice Field Area; | 9. Partial Occlusion by Grass; 10. Partial Occlusion by Humans; 11. Partial Occlusion by Tractor Wheel; 12. Partial Occlusion by Small Irrigation Channel; | 13. Grass; 14. Irrigation Channel; 15. Humans; 16. Small Huts; 17. Houses; 18. Sky (Clouds); 19. Trees. |

As discussed previously, the RIFIS dataset contained images comprising 19 features. In Figure 2, we presented several examples of an RIFIS showing a combination of features, such as different levels of illumination (Figure 2a); strong glare and paddy field conditions (Figure 2b); and small irrigation channels (Figure 2c), partial human occlusion (Figure 2d,e), cloudy afternoons and partially ploughed rice fields that make the sky reflected in puddles and detected as clouds (Figure 2f); fog; foreground objects and city skylines (Figure 2g); huts; pools of water and glare (Figure 2h); and partial occlusion by grass (Figure 2i).

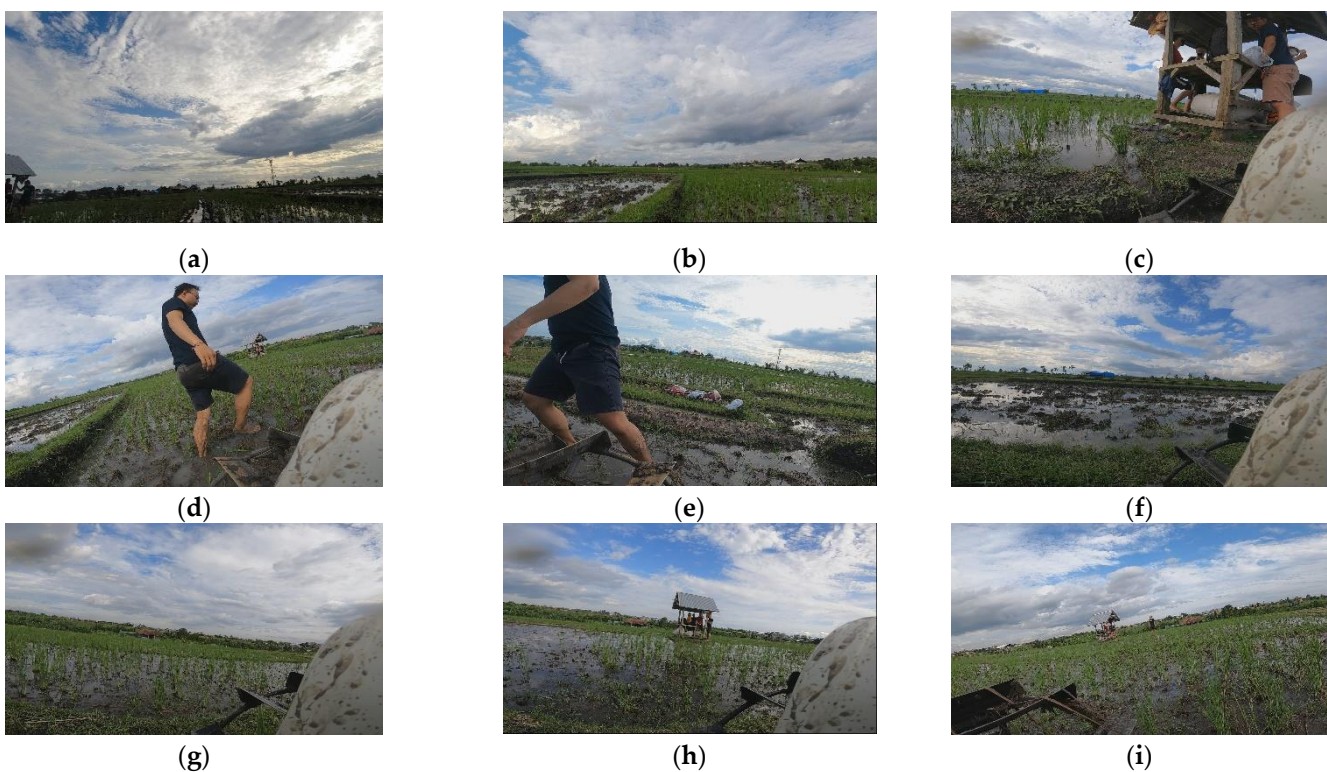

| | | |
|---|---|---|
| (**a**) | (**b**) | (**c**) |
| (**d**) | (**e**) | (**f**) |
| (**g**) | (**h**) | (**i**) |

**Figure 2.** RIFIS dataset showing a combination of features.

The final collection of images was manually annotated using the website-based tool makesense.ai. Annotations had two purposes: first, to identify the RIFIS, and second, as a benchmark for evaluating the RIFIS detection algorithm's performance. We manually drew and labeled sidewalk area polygons for each image. The annotation software outputted a JSON file from which the RIFIS polygon points and recommended ground truth (GT) values were extracted and calculated. Figure 3 depicts the manual annotation procedure using the software makesense.ai

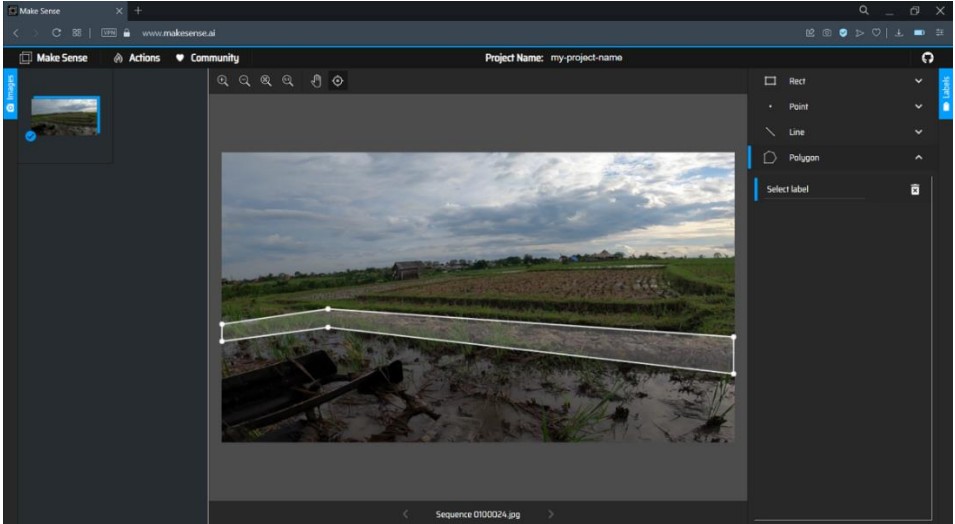

**Figure 3.** Web-based image annotation software (makesense.ai).

The ground truth (GT) value identified the real position of the object of interest within an image. A GT schema depicted in Figure 4 was developed to obtain the rice field sidewalk GT values. There were three GT schemas, namely the RIFIS area, which formed triangular,

square, and concave polygons. The GT schema presented in Figure 4a consisted of eight points forming the RIFIS polygon (sidewalk) area, namely P1 (x1,y1), P2 (x2,y2), P3 (x3,y3), P4 (x4,y4), P5 (x5,y5), P6 (x6,y6), P7 (x7,y7), and P8 (x8,y8). Meanwhile, Figure 4b only had six points, Figure 4c had four points, and Figure 4d had three points. The sidewalk area separated the rice field and outside area.

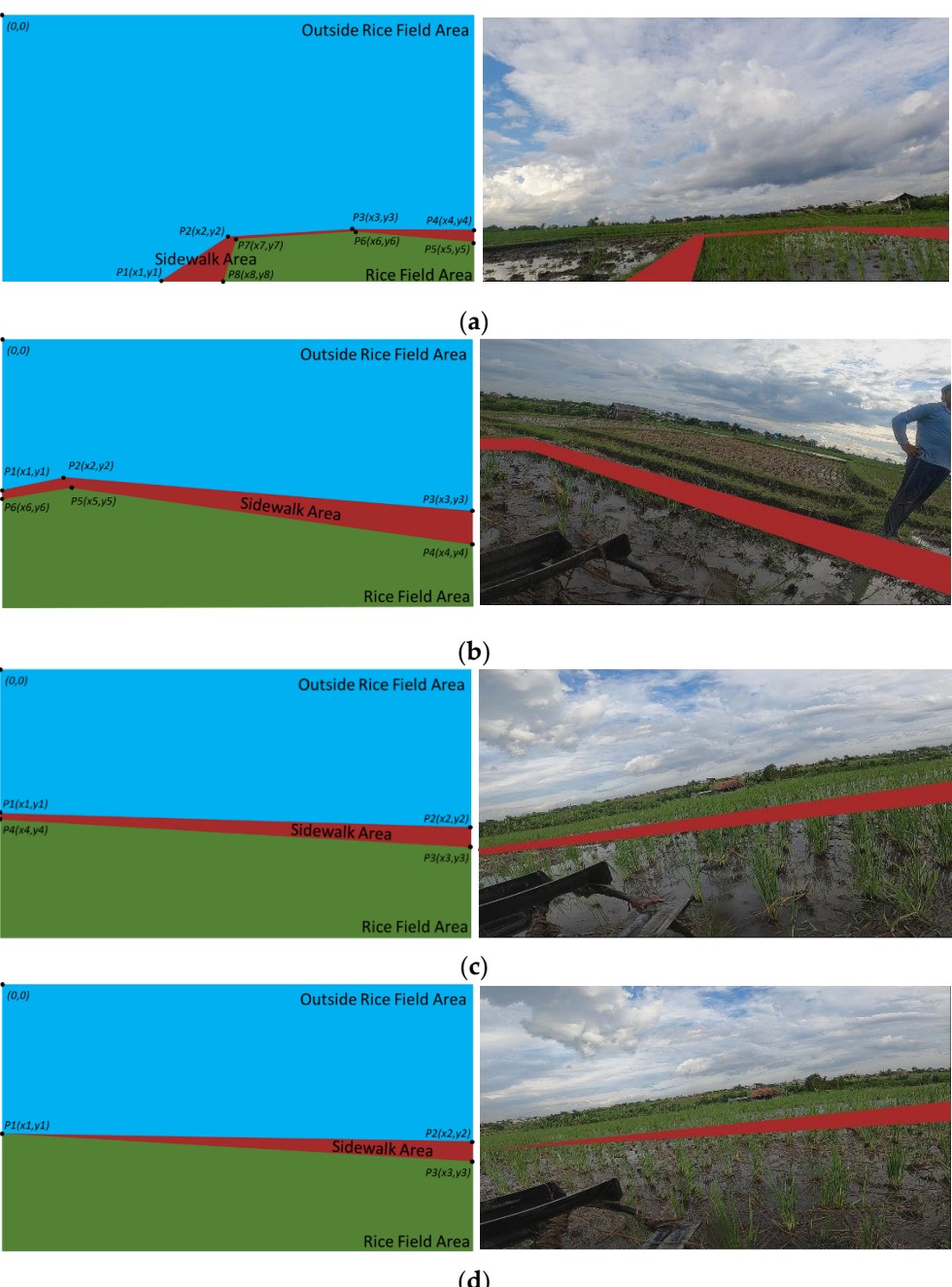

**Figure 4.** Ground truth labeling schema: (**a**) eight points; (**b**) six points; (**c**) four points; (**d**) three points.

Table 3 shows the structure of the RIFIS JSON file as the annotation results, containing two main parts (image and annotation arrays). In our JSON file's annotations field, "id" represented a single image object, "iscrowd" indicated whether the segmentation pertained to a single object or a group/cluster of objects, and "category_id" corresponded to a unique category listed in the categories section. There were two distinct types of labeling: (1) annotation of polygonal segmentation and (2) annotation of rectangular bounding

box. Figure 5 represents examples of image labeling from our RIFIS dataset. As shown in Figure 5a, the polygonal segmentation annotation included a float array segmentation list of vertices (x, y pixel positions). Figure 5b shows the x and y coordinates of the upper left and lower right corner arrays for the rectangular bounding box. "Area" represented the area of the bounding box in each image. Object detection was typically described as detecting a rectangular bounding box and a class label for each object of interest in an image. In instances of segmentation, a pixel-by-pixel segmentation was created for each occurrence. Our target object was the rice field sidewalk, which was not suitable for object detection, segmentation, or depth perception tasks, all of which are required by other systems, such as autonomous or assistance systems. The proposed dataset included a variety of annotations for the sidewalk environment. To the best of our knowledge, this was the first large-scale sidewalk dataset that included annotations for instance-level objects (bounding box and polygon segmentation) and ground-truth depth.

**Table 3.** The structure of the RIFIS JSON file.

| Images [ ] | | Annotations [ ] | |
|---|---|---|---|
| id | integer | id | integer |
| width | integer | iscrowd | Boolean |
| height | integer | image_id | integer |
| file_name | string | category_id | integer |
| | | segmentation | float [ ] |
| | | bbox | float [ ] |
| | | area | float |

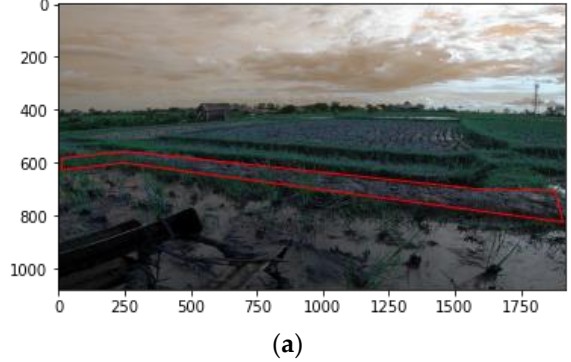 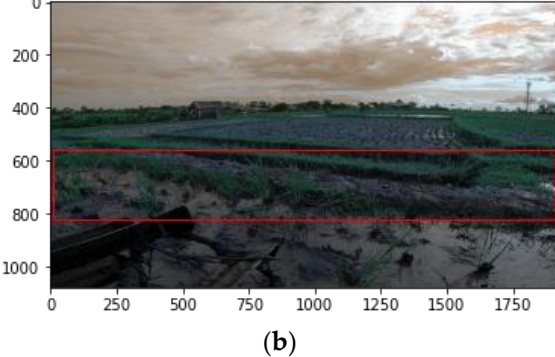

(**a**)       (**b**)

**Figure 5.** Examples of RIFIS dataset showing image labeling: (**a**) polygonal shape; (**b**) rectangular bounding box shape.

### 2.2. Tractor Location and Orientation Dataset

The data obtained through sensors mounted on the tractor were then stored in a database using internet of things technology with the MQTT protocol. The stored data had an index ('id') as the primary key, followed by data on the date that the data were recorded, in the format "YYY-MM-DD HH:MM:SS". Tractor orientation data were obtained from 'yaw', 'pitch', and 'roll' values from the gyroscope sensor; 'x', 'y', 'z' values from the accelerometer sensor; and 'a' (azimuth) values from the compass sensor. The location data of the tractor were recorded using a GPS sensor where the coordinates (longitude and latitude) were the primary reference. The data recorded on the MQTT server were then exported into .sql form to be processed on the local server. The data were then cleaned of noise from GPS reading errors, which were then exported into .xlsx to be more easily analyzed and used. After cleaning, there were a total of 3728 data. Figure 6a shows the electrical component implementation of the data logger; meanwhile, Figure 6b shows the final packaging of the data logger; we used an external antenna to enhance the ESP32 TTGO T-Call signal. The description for this hardware logger can be seen in Table 4.

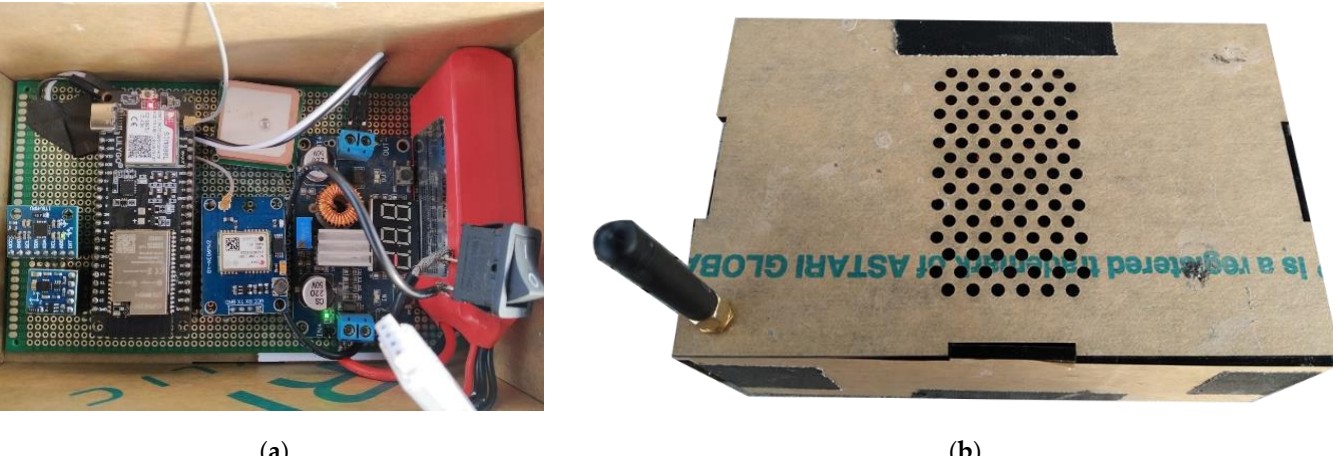

(**a**)                                                    (**b**)

**Figure 6.** Location and orientation logger: (**a**) electrical component assembly; (**b**) final packaging with IoT external antenna.

**Table 4.** Data description captured by the logger device.

| Device | Data Variable | Example Value | Unit |
|---|---|---|---|
| ESP32 TTGO T-Call | Date-Time | 2021-12-21 10:18:06 | yyyy-mm-dd hh:mm:ss |
| Gyroscope | Yaw | −36.219238 | deg/s |
| | Pitch | 2.912616 | deg/s |
| | Roll | −13.965352 | deg/s |
| Accelerometer | X | −39 | m/s$^2$ |
| | Y | −87 | m/s$^2$ |
| | Z | 266 | m/s$^2$ |
| Magnetometer | Azimuth | 284 | deg |
| GPS | Longitude | −8.632576 | deg |
| | Latitude | 115.144852 | deg |

*2.3. Foldering Structure*

The hierarchical folder structure of the RIFIS dataset is shown below:

- RIFIS
  - Images
    - dataset
    - annotations.json
  - LocationOrientation
    - Location-orientation.xlsx
  - Videos
    - FrontCamera
    - LeftCamera
    - RightCamera

## 3. Dataset Acquisition Methods

*3.1. Location and Source of Collection*

Rice field sidewalks are the boundaries of rice fields from one plot to another, usually measuring 30 cm or more. In addition to functioning as a barrier to rice fields, docks, or rice field sidewalks, there are also many functions and uses for farmers. It can reach a width of 1 m or more in certain areas. In some regions, farmers can use rice field sidewalks as

access roads for farming by farmers to transport crops and fertilizers during the fertilization period for rice plants. Routine maintenance of rice field sidewalks is carried out by cleaning them from weeds and sweeping or spraying herbicides. In addition to treating weeds, the barriers must be added with mud and trimmed to keep the rice fields from collapsing.

The rice field is one of the sub-agricultures that provide staple food. Generally, rice fields are used for rice cultivation. However, several stages must be carried out before carrying out the rice planting process, including the process of ploughing the fields. Ploughing is the activity of cultivating the land by turning the soil so that the soil becomes smooth and easy to plant in. The process of ploughing rice fields consists of two processes, namely the process of loosening the soil and the process of refining the soil. The process of loosening the soil currently still uses a tractor. Many tractors are available today, both two- and four-wheeled. In general, the movement of the tractor when carrying out the process of ploughing the fields forms a spiral pattern, as in Figure 7, which was the scenario for collecting RIFIS dataset images in this study. The tractor moved from the start to the finish points with the RIFIS as a barrier. The path that was traversed is called the footprint. We can see a top-view image (using a drone) of the RIFIS image data collection scenario in Figure 8.

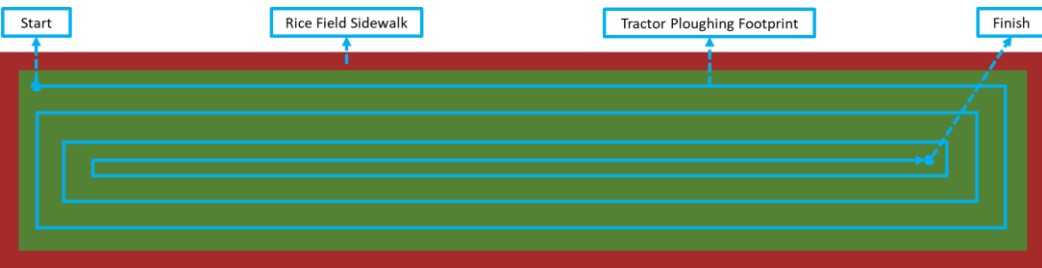

**Figure 7.** The ploughing process using walk-behind hand tractor.

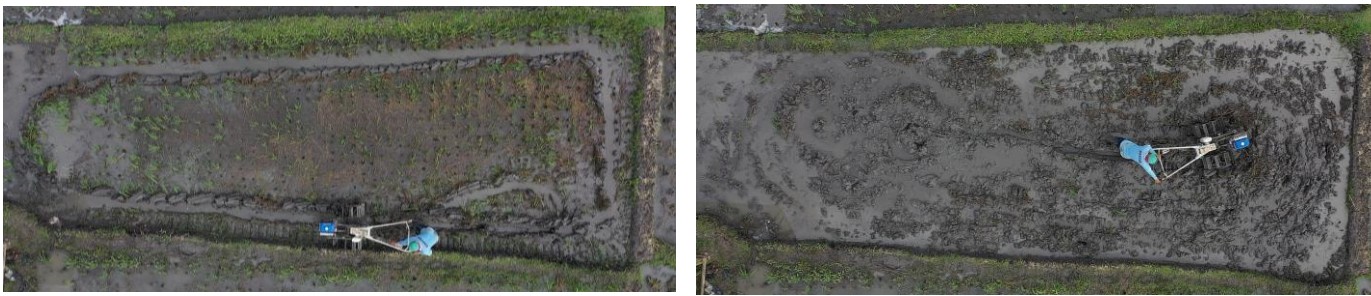

**Figure 8.** Top-view image of the RIFIS image data collection using a drone.

The selection of the observation location was the main factor that affected the dynamics of the features in the RIFIS image. For example, the observation location was in a rice field area where the neighboring rice fields were in a condition where some had been ploughed and some had not. The condition of the cultivated rice fields had similar characteristics to RIFISs, producing dynamic conditions according to reality. Considering this fact, two locations with different longitude and latitude coordinates in Bali, Indonesia were selected for the data collection experiment (Figure 9a). More details about these locations are provided in Figure 9b and Table 5.

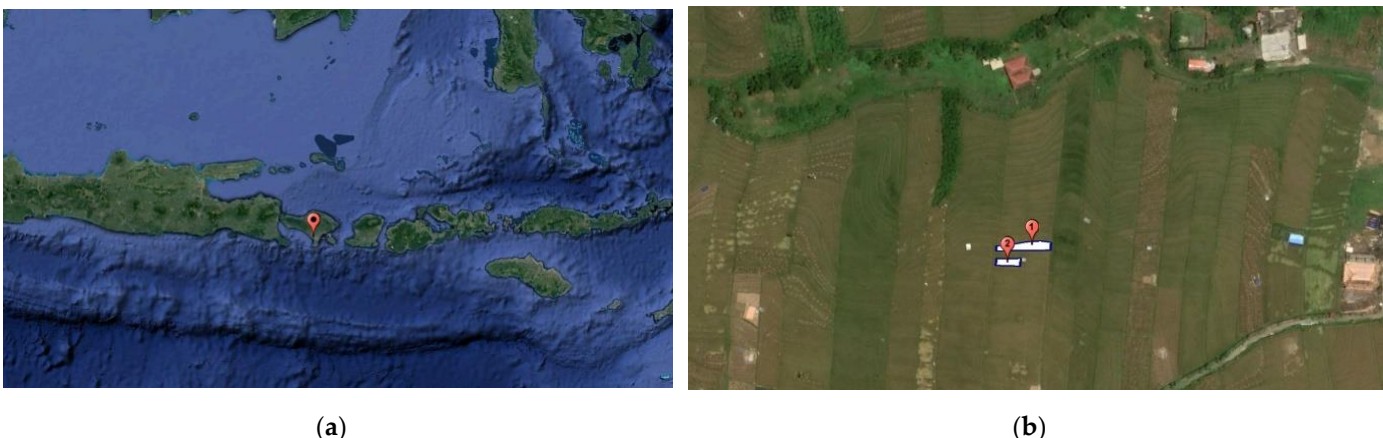

(**a**)          (**b**)

**Figure 9.** Data collection site: (**a**) Bali, Indonesia; (**b**) two locations in Uma Desa Canggu.

**Table 5.** Details of geographical locations for data collection.

| Nature of Location | Location Name | Geographical Coordinates |
| --- | --- | --- |
| Rice Field 1 | Uma Desa Canggu | −8.632394°; 115.144956° |
| Rice Field 2 | Uma Desa Canggu | −8.632368°; 115.144836° |

### 3.2. Camera and Recording Support

To capture RIFIS images in the process of ploughing fields, we used a GoPro Hero 9 camera. The camera settings used were auto (zoom 1.0×) with an image resolution of 1920 × 1080 and a 60 frames per second (fps) frame rate. The lens setting used in our research was wide, with an ISO in the minimum value range of 100 to a maximum of 6400. The three cameras were mounted on the top of the front of the tractor. The first camera faced the right diagonal, the second camera faced forward, and the third camera faced the left diagonal. The camera placement on the tractor can be seen in Figure 10. In Figure 11, we can see the results of the captures of the three cameras. We recorded all video sequences of the dataset by placing the camera on top of a tractor, ploughing a field with three different shots (diagonal left, front, and right). Three sets of footage were taken with an above-shot camera angle relative to the RIFIS.

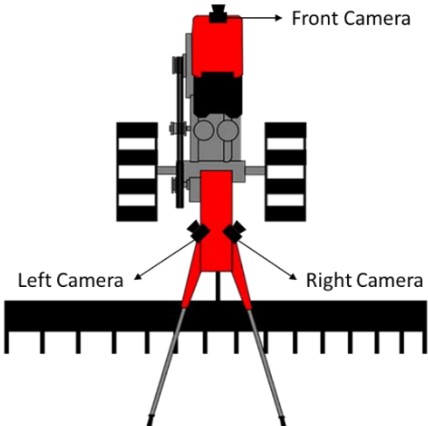

**Figure 10.** The camera placement on the tractor.

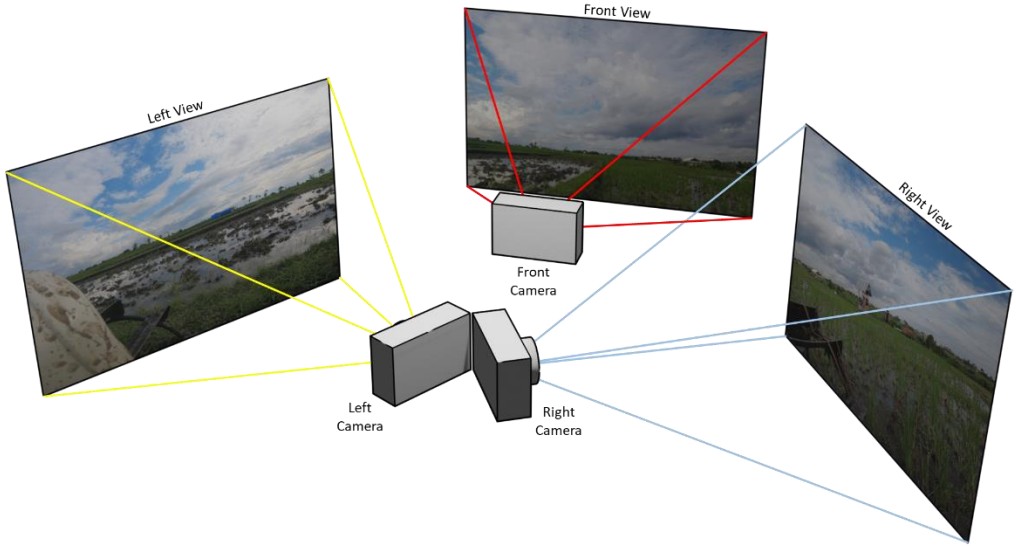

**Figure 11.** Capture results from three cameras.

### 3.3. GPS, MPU, and Compass

The location and orientation data of the tractor were recorded to view and analyze the movement patterns as supplementary data. A set of hardware was embedded in the tractor to achieve this goal. IoT technology with the MQTT protocol was used as a liaison between the hardware and the server. ESP32 Lilygo T-Call 1.4 is a microcontroller equipped with a SIM800L module. This allowed it to communicate over the internet without needing a separate access point module [27]. Three sensors were used to obtain tractor movement data, namely the U-Blox Neo-6M as a GPS module to obtain location data for longitude and latitude coordinates. To obtain tractor orientation data, an MPU6050 GY-521 was used as the gyroscope–accelerometer sensor and a GY-271 as the compass sensor. For this study, a 1-s interval was used to record all the location and orientation data of the tractor. Figure 12 illustrates the wiring in the three sensors and microcontroller diagrams during the data collection experiment.

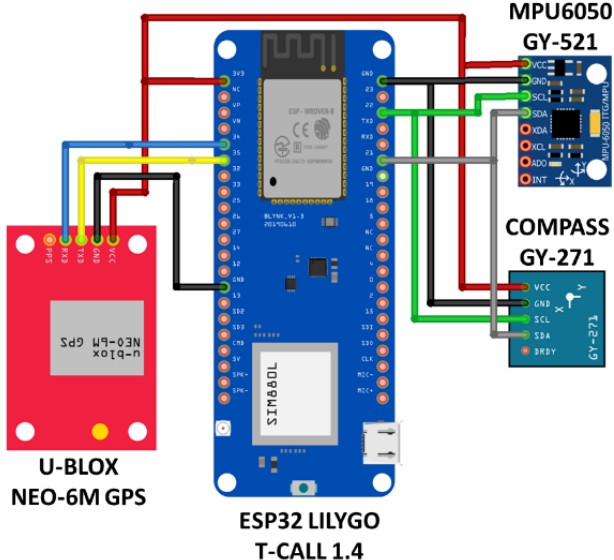

**Figure 12.** Data logger wiring diagram.

## 4. Dataset Evaluation

### 4.1. Mask R-CNN

An evaluation was carried out to validate this dataset using the Mask region-based convolutional neural network (Mask-RCNN) method. This method is an instance of segmentation consisting of object detection and semantic segmentation. Mask R-CNN is a continuation of Faster RCNN [28,29], which focuses on object detection by providing a region of interest (RoI) bounding box along with its label. Meanwhile, the fully convolutional network is a masking technique used to handle semantic segmentation in Faster RCNN [30]. Figure 13 shows the Mask-RCNN network architecture used to evaluate the RIFIS image dataset; the entire model in this study was built on this architecture. This model can classify objects and assign bounding boxes and masks to the detected objects. The calculation of the multi-mask loss function can be seen in Equation (1), with detailed calculations in Equations (2)–(4) [30]. In Table 6, the detailed definition of each symbol used is shown.

$$L = L_{class} + L_{box} + L_{mask} \tag{1}$$

$$L_{class} + L_{box} = \frac{1}{N_{cls}} \sum_i L_{cls}(p_i, p_i^*) + \frac{1}{N_{box}} \sum_i p_i^* \, L_1^{smooth}(t_i, t_i^*) \tag{2}$$

$$L_{cls}(\{p_i, p_i^*\}) = p_i \log p_i^* - (1 - p_i^*) \log(1 - p_i^*) \tag{3}$$

$$L_{mask} = \frac{1}{m^2} \sum_{1 \le i,j \ge m} \left[ y_{ij} \log \bar{y}_{ij}^k + (1 - y_{ij}) \log \left( 1 - \bar{y}_{ij}^k \right) \right] \tag{4}$$

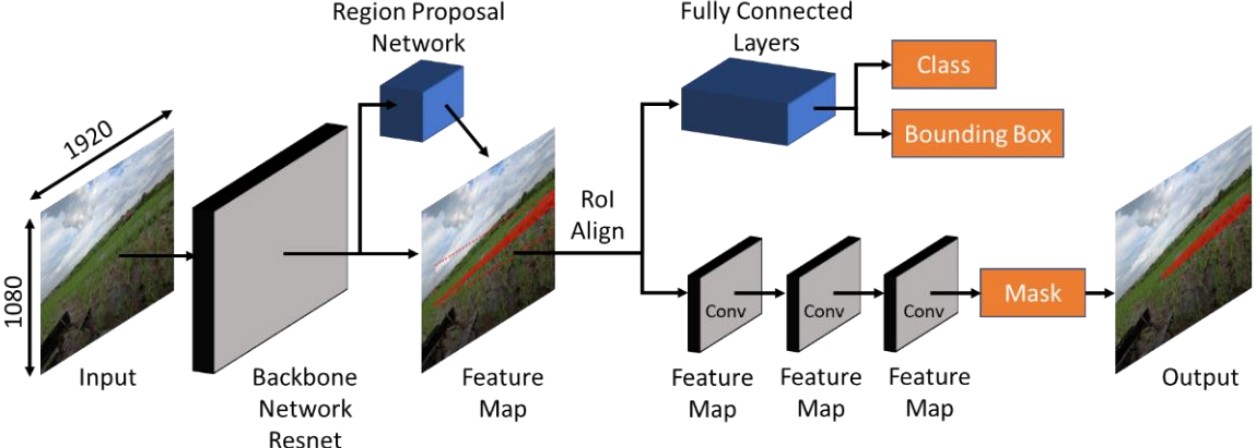

**Figure 13.** Mask-RCNN network architecture.

**Table 6.** Nomenclature.

| Symbol | Definition | Symbol | Definition |
|--------|------------|--------|------------|
| $i$ | the index of an anchor | $p_i^*$ | ground truth label |
| $L$ | loss function | $t_i$ | predicted four parameterized |
| $L_{class}$ | classification loss | $t_i^*$ | coordinates of the bounding box |
| $L_{box}$ | bounding box regression loss | $N_{cls}$ | mini-batch size |
| $L_{mask}$ | mask prediction loss | $N_{box}$ | number of anchor locations |
| $p_i$ | predicted probability of anchor $i$ as RIFIS | | |

### 4.2. Dataset Evaluation Results

All setups were implemented in Google Collaboratory Integrated Development Environment (IDE) (Colab) using NVIDIA-SMI 460.32.03 GPU, Tesla K80 28GB with driver version 460.32.03, and CUDA version 11.2. The RIFIS dataset was installed into Google Colab using Google Drive in the form of a JSON file. All algorithms were developed using

Python programming language. Experiments on the deep learning model were conducted to detect the rice field sidewalk. Mask R-CNN was trained for five epochs with 500 steps each (we used the basic detector setup without modification). The model was trained for about 4 h with 863 images for training and 107 for testing.

The performance assessment of the methods is tabulated in Table 7. Train Loss is the output value on the training data. Generally, the smaller the loss value, the better the results; this was our reference in evaluating the networks and datasets [24]. Meanwhile, the Validation Loss is the output value on the validation data. Based on the fifth epoch, the Train Loss was ~0.25, which was slightly lower than the Validation Loss, which was ~0.27. This showed that the network was overfitting because its performance was worse on data that had never been seen before. This problem can be overcome in the future by modifying the model by increasing the layer of neurons [24]. Based on the data from the first and last epochs, the network increased during the training from a loss of ~0.88 to a smaller loss of ~0.25. Figure 14 shows the Mask-RCNN sidewalk mask visualization. Figure 15 shows the sidewalk detection results using Mask-RCNN. The detection accuracy of the model was higher if there was only one sidewalk. Based on this, the created dataset could provide images and annotations that could be used for RIFIS detection.

**Table 7.** Train and Validation Loss Values.

| Name | Number of Steps | Time | Train Loss | Validation Loss |
|---|---|---|---|---|
| Epoch 1 | 500 | 847 s | 0.8881 | 0.3588 |
| Epoch 2 | 500 | 453 s | 0.4238 | 0.3005 |
| Epoch 3 | 500 | 455 s | 0.3400 | 0.3516 |
| Epoch 4 | 500 | 454 s | 0.2948 | 0.3902 |
| Epoch 5 | 500 | 455 s | 0.2510 | 0.2757 |

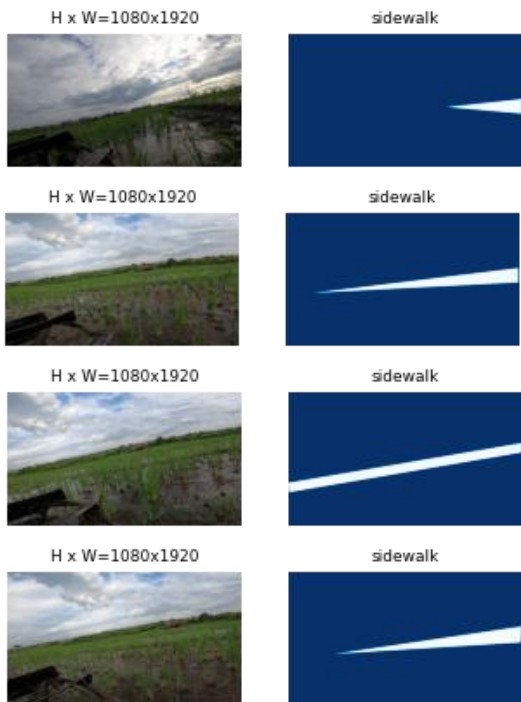

**Figure 14.** Example of sidewalk mask (white area).

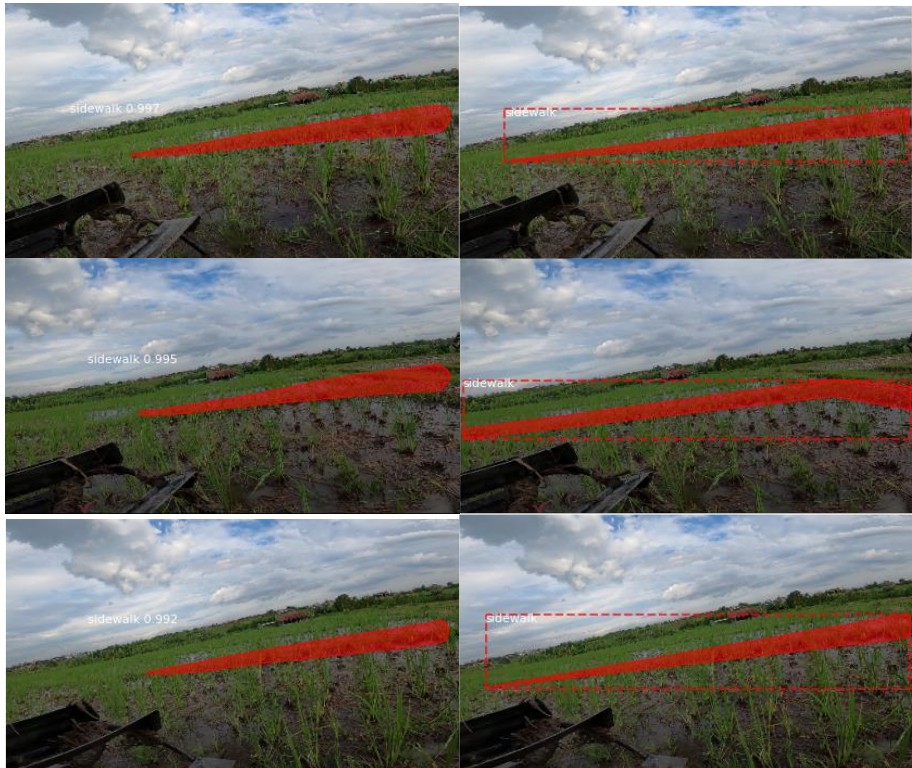

**Figure 15.** Sidewalk detection results.

## 5. Conclusions

In this study, we introduced a novel, comprehensive, and diverse dataset called the RIFIS dataset to allow the researchers to develop the process of automation of ploughing fields using hand tractors. The RIFIS dataset contained 3723 images, 18 videos, and a JSON file with polygonal and bounding box labeling values for 970 images. The RIFIS dataset could automate the ploughing of rice fields not just at the time of rice planting but also at the time of rice harvest, as well as for a variety of other purposes throughout the year. This was the first ever compilation of rice field sidewalk annotations. The RIFIS enabled the training of deep learning models for sidewalk detection in paddy fields. To assess the quality of the RIFIS dataset, a Mask-RCNN model was employed to develop a preliminary sidewalk detection algorithm. It was projected to improve the fine-grained segmentation of sidewalk site discoveries and reduce false positives and negatives for deep learning models. As supplementary data, the tractor location and orientation excel files were included with 'yaw', 'pitch', and 'roll' values obtained from the gyroscope sensor; 'x', 'y', and 'z' values from the accelerometer sensor; and 'a' (azimuth) values from the compass sensor and the location of the tractor from the GPS sensor. This allowed the researchers to examine the movement patterns of the tractor. The main goal of our RIFIS dataset was that the research and models based on the RIFIS dataset could be used for sidewalk detection, distance prediction, tractor location, and orientation tracking to build an innovative tractor autonomous control system.

This study had two significant limitations that could be addressed in future research. First, the RIFIS dataset was exclusively collected from paddy fields in Indonesia, Bali. Second, this research was limited to collecting images, videos, and annotations of paddy field sidewalks. Further research on integrating camera detection results and sensor readings is still needed. As a future development, sidewalk detection results using Mask R-CNN can be combined with basic image processing and detecting the distance between the lower center point of the image and the generated mask. The basic concepts of further research that can be developed can be seen in Figure 16. This method can be implemented on all three cameras and then combined with the reading of several sensors to decide the

tractor's movement. In the future, we aim to make more in-depth comparisons to more precisely detect the sidewalk's location and automate cultivating rice fields using hand tractors.

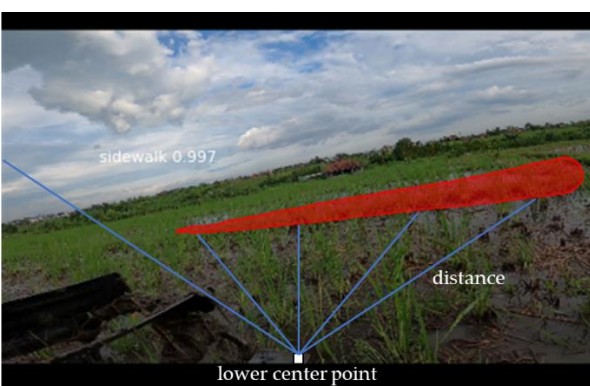

**Figure 16.** Future research.

**Author Contributions:** Conceptualization, P.N.C. and D.M.; methodology, P.N.C. and D.M.; software, P.N.C. and D.M.; validation, P.N.C. and D.M.; formal analysis, P.N.C. and D.M.; investigation, P.N.C. and D.M.; resources, P.N.C. and D.M.; data curation, P.N.C. and D.M.; writing—original draft preparation, P.N.C.; writing—review and editing, P.N.C. and D.M.; visualization, P.N.C.; supervision, D.M.; project administration, D.M. All authors have read and agreed to the published version of the manuscript.

**Funding:** This research received no external funding.

**Institutional Review Board Statement:** Not applicable.

**Informed Consent Statement:** Not applicable.

**Data Availability Statement:** The data presented in this study are openly available in https://doi.org/10.21227/pnxx-3t40 (accessed on 14 August 2022) with doi: https://doi.org/10.21227/pnxx-3t40 (accessed on 14 August 2022).

**Acknowledgments:** We would like to express our deepest gratitude to the Ministry of Research, Technology and Higher Education of the Republic of Indonesia, the STIKOM Bali Institute of Technology, and the Business and Rajamangala University of Technology Thanyaburi (RMUTT) for the support and facilities that were provided for this research.

**Conflicts of Interest:** The authors declare no conflict of interest.

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
