# Peer review of "RIFIS: A Novel Rice Field Sidewalk Detection Dataset for Walk-Behind Hand Tractor"

_data, 2022_

Round 1

Reviewer 1 Report

Author main contribution is to automation of ploughing field using hand tractors using RIFIS video dataset and use deep learning model for sidewalk detection in paddy fields. The concept is very good. the following points observed which has to incorporate-

-all 19 features values of dataset not cleared like 2 & 18 and what hypothesis taken to select only 19 features not cleared

- epochs increases, what is impact of proposed model outcome in context to validation & train loss

-complete manuscript (tables, figures) is in MDPI standard format 

Reviewer 2 Report

1. The authors have made a good effort by detecting the rice fields with centric data. But how the proposed method is integrated with Hardware setup?

2. No clear data is provided. A table form data can be included.

3.  Separate sub section on background with current research article can be included and cited

4. Conclusions must be greatly improved with end result description

5. Introduction lacks major description of proposed methodology.

6. Objectives must be framed as separate subsection.

7. Overall paper must be reorganized with appropriate sections.

8. If the authors are proving a new model then exact comparison cases can be added.

Reviewer 3 Report

Authors are suggested to improve the topological error. Results can be interpreted by different fold validation techniques. A comparative literature survey is required also comparative results to understand a research work easily.

Reviewer 4 Report

1. The title is straightforward and reflects the totality of the study. However, the Introduction is not well written. The problem being solved is not clear. 

2.  Based on Table 1, only afternoon day condition characterizes the RIFIS dataset. What is the reason why early morning or noon time were not considered? How the inclusion of early morning and noon will impact the Gaussianity of RIFIS?

3. In Figure 4, it would be better if the color coded ground truth is partnered with corresponding raw image. In that case the readers could understand the significance of GT labelling. 

4. Why polygonal and rectangular-shaped bounding boxes have been employed? 

5. The manuscript is missing of technical discussions of the results. Even the advantages of RIFIS was not highlighted. It is recommended that the authors compare their work to other studies by making a summary table. 

6. Other corrections: 

Line 19: "1920 1080 pixels" to "1920 x 1080 pixels"

Line 171: Different font face

Round 2

Reviewer 2 Report

The authors have incorporated the recommended suggestions. Hence it shall be published in its current form.